# Numerical Designing of Fiber Reinforced Concrete Eco-Constructions

**DOI:** 10.3390/ma16072576

**Published:** 2023-03-24

**Authors:** Pierre Rossi

**Affiliations:** MAST-EMGCU, University Gustave Eiffel, IFSTTAR, F-77447 Marne-la-Vallée, France; pierre.rossi@univ-eiffel.fr

**Keywords:** eco-constructions, numerical models, fiber reinforced concretes, carbon footprint, cracking process

## Abstract

This paper focuses on the use of numerical tools, as a finite elements method, to conceive fiber reinforced concrete (FRC) eco-constructions. It highlights the fact that these are the most suitable tools (much more than the Eurocodes, for example) to predict the cracking process of FRC constructions at their service limit state and, therefore, to predict their durability. Following a critical analysis of the existing finite element models for FRC cracking, it describes in more detail a probabilistic one. This model appears very suitable for providing precise information about crack openings that are inferior or equal to 300 microns. Finally, it presents an example of the use of this numerical model to optimize an FRC track slab in order to reduce its carbon footprint. This study, although partial and incomplete, shows that the best way to reduce the carbon footprint of this type of construction is to reduce its thickness.

## 1. Introduction

Eco-designing concrete construction is not only about decreasing the amount of cement used in the concrete mix design. It is also very important to optimize the geometry and dimensions of the construction. To do this, it is necessary to use powerful design tools.

From this perspective, the improvement of the durability of concrete constructions is a very important objective.

The use of Fiber Reinforced Concretes (FRC, including Ultra-High-Performance Fiber Reinforced Concretes, UHPFRC) permits this improvement. This is due to the fact that fibers are very efficient (more than the usual steel rebars) at controlling crack openings inferior or equal to 300 microns [1]. This means that FRCs are very efficient during the service limit state of the concrete constructions. However, they also have the disadvantage of containing a lot more cement per cubic meter. Consequently, it is imperative to use, in construction, the least possible amount of these materials while ensuring the intended functions. It is therefore essential that the dimensioning method considers the increased mechanical and physico-chemical performance of the material as accurately as possible.

The Eurocodes, in Europe, and other standards in the world that are commonly used for concrete construction design, are incapable of favourably considering the cracking process of FRCs, especially their cracking process at the service limit state.

The finite element method appears to be a solution for use in future to meet this challenge.

This paper proposes, after a critical analysis of the existing non-linear finite elements models used for FRC, to focus on the best method for providing relevant information concerning crack opening in FRC constructions when their service limit state is concerned.

It also proposes an example use of this numerical model for optimizing FRC concrete construction in order to reduce its carbon footprint. 

## 2. Numerical Models for Cracking in FRC

As the objective of this paper is to consider the problem of durability in FRC constructions associated with the service limit state of these constructions, the models presented in this chapter are compared with each other, considering their performance with respect to their ability to predict crack openings less than or equal to 300 microns.

The other choice, in the framework of this critical analysis, is to consider only models relevant for performing simulations of real structures.

In the literature, there exists two main model families for analyzing the cracking process of FRC structures: the diffused cracking models and the explicit cracking models.

It is important to focus on the fact that, for all of these numerical models, the main mechanical property of FRC used is its behavior in uniaxial tension and, more especially, the mechanical relationship between the uniaxial tensile strength and a crack opening.

This relationship can be obtained directly by performing a uniaxial tensile test on notched specimens [1]; or, indirectly, by performing bending tests on small beams or on small slabs and by using inverse approaches [2,3].

The first type of test is the best one for directly and precisely obtaining the mechanical data; however, this test is considered difficult to perform. This is the reason why many researchers prefer to perform the second family of tests even though these tests are indirect and less precise.

### 2.1. Diffused Cracking Models

These models do not consider cracks as real cinematic discontinuities but as microcracked zones in which the density of the microcracks continuously increases until they create a kind of hole (no more stresses are transmitted). Numerically speaking, in the framework of finite elements theory, the microcracked zones are modeled by using volume elements. This means that the volume of the holes corresponds to the volume elements adopted.

In these models, which are generally determinist, the non-linear mechanical behavior (related to the microcracking process) of the volume elements is linked to the post-peak behavior in the tension of the FRC. This post-peak behavior is generally characterized by two material parameters:The shape of the post-peak (this means after the linear elastic part of the tensile behavior of the FRC) behavior of the tensile stress-strain curve.The post-peak energy dissipation related to the tensile stress-strain curve. This dissipation energy is very often called G_f_.

To summarize, the diffuse cracking models, applied to FRC, are based on the use of the non-linear finite elements method with volume elements. The non-linear behavior considered is that of the FRC in tension, in the form of a relationship between the tensile stress and the tensile strain.

Experimentally speaking, the post-peak behavior of FRCs corresponds to the post-localization cracking; this means after it progresses from diffused microcracking to a localized macrocrack [4]. Then, it clearly appears that the diffuse cracking models are not physically based.

The post-peak behavior in tension, used as the main mechanical characteristic in the diffused cracking models, comes from the experimental tensile stress-crack opening curve (see Section 2). To pass from the experimental tensile stress-crack opening curve to the theoretical tensile stress–tensile strain used in the model requires the introduction of a length that permits the division of the crack’s opening value to obtain the “equivalent” tensile strain.

In the majority of the diffused cracking models, this length is linked to the volume element size of the finite element mesh.

This length chosen and used in the numerical simulation is very important because it permits, during the simulation, the transformation of the non-linear strains observed in crack openings.

These models, which at the present time are the most commonly described in the literature and used in practice, have a large drawback where the determination of crack openings that are less or equal to 300 microns is concerned.

Indeed, the diffused cracking models are not physically based because they “transform” a localized crack in a diffused microcracking zone. This strong physical approximation leads to a strong approximation of the cracking pattern of a given FRC structure being obtained, especially at the service limit state. The cracks are always more numerous and less spaced out at the service limit state than at the ultimate limit state. Being that the G_f_ values are a lot larger for FRCs than for normal concretes, the diffused cracking models result in a greater spread of the damaged (microcracked) zones and an underestimation of crack openings. Therefore, these models do not lead to a fairly reliable and secure response to the durability of FRC structures.

However, at the ultimate limit state, whereby cracks are localized with larger openings, the diffused cracking models are less approximate.

The more well-known diffused cracking models for FRC are damaged models [5,6,7] and smeared crack models [8,9,10,11], which are mechanically equivalent.

It can be noted that the smeared crack model developed at Ecole Polytechnique of Montréal (Canada) [10,11] provides a better evaluation of crack openings in FRCs than other diffused cracking models. This is due to the fact that it uses an explicit algorithm of resolution while the other models use an implicit algorithm of resolution. This choice leads to better localization of the cracks. Despite this improvement, and even though the evaluation of crack openings at the ultimate service state is better, it has not been demonstrated to be enough to obtain a relevant evaluation of crack openings at the service limit state.

### 2.2. Explicit Cracking Models

Contrary to the diffused cracking models, the explicit cracking models have the objective of treating the cracks as “real” cracks, which means as cinematic discontinuities and not as non-linear strains.

There mainly exists two families of explicit cracking models that can analyze cracking at the scale of the structure: the cohesive crack models and the probabilistic explicit cracking models.

#### 2.2.1. The Cohesive Crack Models

These models were the first to be used to analyze crack propagation in FRCs [12,13,14,15].

In these models, the propagation of existing cracks is modelled by using interface elements. These interface elements are located at the front tip of the existing cracks.

The interface elements are interesting because their behavior (linear or non-linear) is not considered via a stress–strain relationship but via a stress–displacement relationship. This means that the experimental post-peak behavior of the FRC linking the tensile stress to the crack opening can be directly used as the non-linear behavior of the interface elements. This is a large advantage compared to the diffused cracking models. Indeed, the fact that these models are more physically based allows them to avoid the use of a “numerical length”, which is not physically based (so leads to some approximations), to transform a strain in a displacement (so in a kind of equivalent crack opening).

There is nevertheless a large problem concerning the field of utility and therefore of the use of these models.

Indeed, as mentioned before, they are models related to crack propagation and not to crack initiation. This means that it is important to know the initial position of the cracks to study their propagation by using the numerical model. This is not very effective when the structure has many cracks at the service limit state.

So, these models are more effective at following one crack propagation until the rupture of the FRC structure.

#### 2.2.2. The Probabilistic Explicit Cracking Model

The probabilistic explicit cracking model was developed to analyze the cracking process of concrete [16,17]. The numerical model is deeply detailed in [16,17]. It is based on the following physical assumptions:The material being considered is heterogeneous and its local mechanical characteristics are randomly distributed.These mechanical characteristics are scale effect dependent. This means that they depend on the volume of the material considered [18].

From the mechanical and numerical points of view, this can be summarized as follows:Each volume element of the mesh represents a volume of the heterogeneous material.The tensile and shear strengths are distributed randomly over all elements of the mesh [16,17].Cracks are simulated by using non-linear interface elements (quadratic elements). Failure criteria of Rankin in tension and Tresca in shear are used. Once one of these failure criteria is reached, the interface element is considered as opened, simulating a crack’s creation. The tensile and shear strengths, as well as the normal and tangential stiffness values of the element, are then set equal to zero.

The complete bridging action of fibers inside the cracks is simulated as follows [19]:The normal and tangential stresses in the interface element linearly increase with the normal and tangential displacements to simulate the elastic bridging action.This elastic action of the fibers exists until a threshold value related to the normal displacement is reached. From this threshold value, the normal stress linearly decreases with the normal displacement. This is to consider the damage of the bond between the concrete and the fiber. This decreasing evolution is considered through a damage model.Finally, when the normal displacement reaches a second threshold, the action of the fibers is considered negligible, and so, the interface element is considered definitively broken. Its normal and tangential rigidities are then set to zero.The post-cracking energy, dissipated during all of the bridging action of the fibers, is randomly distributed over the finite element mesh. This random distribution is a log-normal distribution function with a mean value independent of the mesh elements’ size and a standard deviation that increases as the mesh elements’ size decreases. This choice is in perfect agreement with the experimental results [20].

To analyze the cracking process in a real FRC structure, the way to determine the values of the model parameters of the fibers’ action is as follows:The mean value is determined from the experimental results related to the uniaxial tensile tests on notched specimens (see at the beginning of Section 2).The standard deviation is determined by an inverse analysis approach. As the mean value of the post-cracking energy is known, the inverse approach consists of simulating the uniaxial tests with different element mesh sizes. For each element mesh size, several numerical simulations are performed. The standard deviation related to each mesh size is the one that best fits the experimental results (from the uniaxial tensile tests). This inverse approach allows for the determination of the relationship between the standard deviation and the finite element mesh size.The two threshold parameters, evocated above, are also obtained by performing an inverse approach. This consists of fitting the post-cracking behavior of the uniaxial tensile tests by using the simplified triangular stress-displacement curve of the model.

It is important to consider that the extension of the probabilistic explicit cracking model to FRC was validated for different structural problems [19,21,22]. These validations were focused on the relevancy of the model to determine crack spacings and openings even for crack openings less than 300 microns.

The model is schematically summarized in Figure 1.

The probabilistic explicit cracking model has been recently used for optimizing an FRC track slab in order to reduce its carbon footprint [23].

This example of use is presented in the following chapter.

## 3. Design Optimization of an FRC Track Slab

All of the study, briefly presented in this chapter, was presented in detail at an international conference [23].

Some years ago, a new concept for railway tracks, called New Ballastless Track (NBT), was proposed [24]. This concept, based on two superimposed independent layers of concrete slabs (i.e., foundation and track slabs), was validated on a real-size mockup [24,25,26].

Figure 2 presents the geometry of the experimental mock-up. The track slab is made of reinforced concrete, called BC5 and the foundation slab is made of plain concrete, called BC3. The mechanical reaction of the ground is considered by using an elastic layer.

The first numerical study [27] was performed to evaluate the possibility of using steel fiber reinforced concrete to replace the original reinforced concrete layer BC5.

It was demonstrated that the use of 78 kg/m^3^ of fibers is mechanically more efficient than the usual ratio of rebars.

It was also indicated that the use of fibers did not exclude the use of some local rebars in zones of high concentration of tensile stresses (this means in the track layer, B3, at the vertical of the joints of the foundation slab, B5).

After this first study, a second one was performed with the objective of optimizing the design of the FRC railway track to decrease its carbon footprint.

The following strategy of optimization was followed:

*Step 1*—Different dosages of fibers from 40 to 78 kg/m^3^, different sectional percentages of bottom rebars from 0 to 8.8%, and different thicknesses of the upper track layer (B3) from 15 to 24 cm were considered in the numerical simulations. Rebars, with a length of 40 cm, were used as in the first study [27]. As requested by the industrial team, only the technical solutions leading to crack openings less than or equal to 100 µm were retained at this step. These correspond to a service limit state design.

As the numerical model being used was a probabilistic one, between 10 and 30 simulations were performed for each technical solution studied.

*Step 2*—Carbon footprint evaluation was performed for the technical solutions resulting from step 1.

All of the technical solutions simulated during the first step of optimization are presented in Table 1.

Table 2 presents only the carbon footprint evaluation related to the technical solutions considered acceptable at the end of step 1. They are expressed in terms of the crack openings (mean and standard deviation) and the number of cracks for the three zones of the loaded BC5 slab (left, central and right zones, see Figure 3).

Figure 3 shows an example of a crack pattern obtained with technical solution n°15 (Table 1)

### Carbon Footprint Evaluations

A comparison of the carbon footprint evaluations concerning the solution of railway tracks reinforced only with rebars [23], and the solutions that used a mix of steel fibers and rebars (Table 3) was made. This comparison used some data from the literature [28].

Table 4 presents the carbon footprint evaluations relating to the various technical solutions. The CO_2_ amounts are calculated for one slab. The gain in CO_2_ emitted was expressed here in comparison with the classical solution (the solution with only rebars). The minus sign denotes a reduction in CO_2_ emission.

Table 4 shows that the best solution for reducing CO_2_ emission is to reduce the volume of concrete used. A small reduction in the slab’s height of 2 cm leads to an almost 10% reduction in CO_2_ per slab. This represents around 5.2 t of CO_2_.

With this idea of reducing the concrete’s volume, solution number 20 becomes a lot more innovative and interesting. It leads to a 37% reduction in CO_2_ emission.

Concerning solution number 20, it would be important, in future, to optimize it by reducing the percentage of steel rebars that also appears very important (surely too much).

## 4. Conclusions

This paper highlights that numerical models are the most suitable tools (much more than the Eurocodes, for example) for predicting the cracking process of FRC constructions at their service limit state. It describes, briefly, the probabilistic explicit cracking model that appears very suitable for providing precise information about crack openings that are inferior or equal to 300 microns. Then, it presents an example of the use of this numerical model to optimize an FRC track slab, which was designed at the service limit state, in order to reduce its carbon footprint. This study shows that the best way to reduce the carbon footprint of this type of construction is to reduce its thickness.

To conclude, this study, although very partial and incomplete, confirms the great potential of numerical models for optimizing the design of eco-constructions.

## Figures and Tables

**Figure 1 materials-16-02576-f001:**
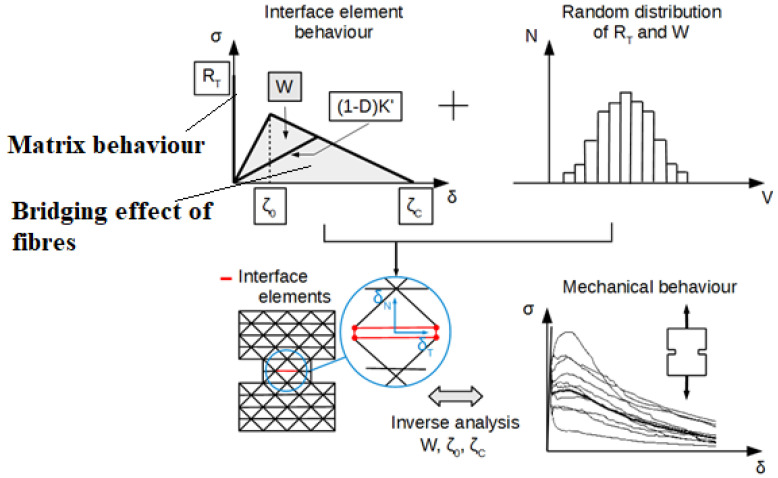
Probabilistic explicit cracking model for FRC.

**Figure 2 materials-16-02576-f002:**
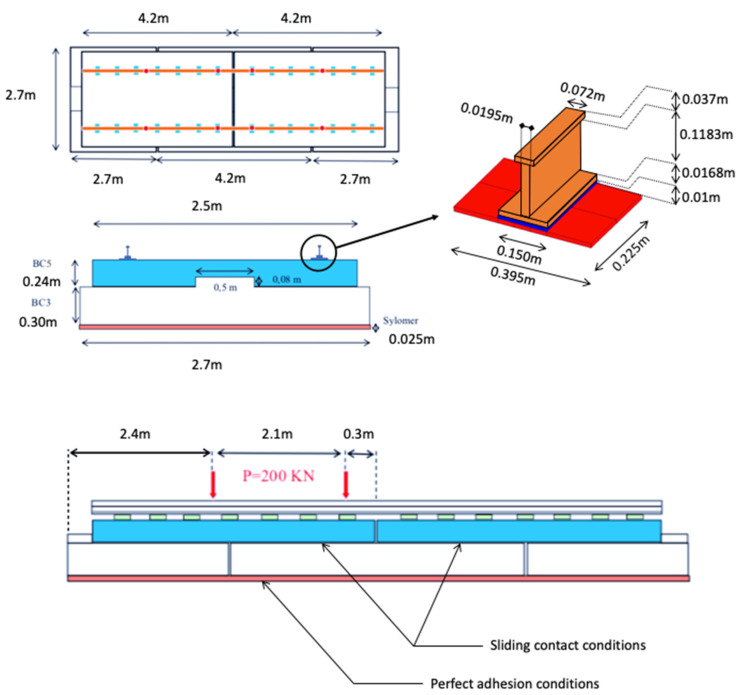
Geometry and dimensions of the Railway Track Mockup.

**Figure 3 materials-16-02576-f003:**
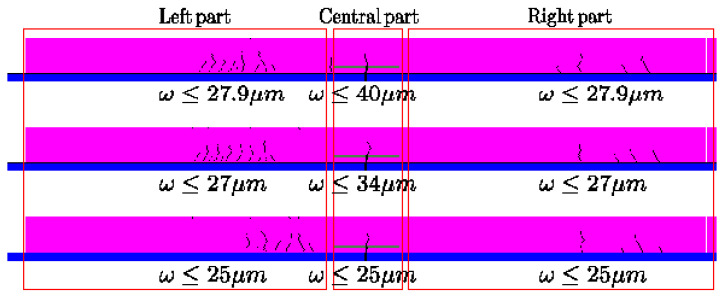
Example of cracking patterns related to technical solution n°15—3 different simulations.

**Table 1 materials-16-02576-t001:** All technical solutions studied (step 1).

Technical Solution Number	Concrete Slab Thickness (cm)	Steel Reinforcement (%)	Fiber Content (kg/m^3^)
1	15	2.2	72
2	15	6.61	80
3	17	5.29	56
4	19.5	0	56
5	19.5	2.2	48
6	19.5	8.81	40
7	22	7.93	76
8	24	4.41	48
9	24	4.41	64
10	17	4.41	68
11	17	2.64	64
12	19.5	7.05	60
13	22	6.17	56
14	22	5.29	64
15	22	5.29	76
16	22	7.93	64
17	19.5	5.29	76
18	19.5	7.93	64
19	24	8.81	80
20	15	17.6	72

**Table 2 materials-16-02576-t002:** Results related to the acceptable technical solutions.

Tech. Sol. Num.	Left Part			Central Part			Right Part		
	Crack Opening (µm)		Crack Number (Mean)	Crack Opening (µm)		Crack Number (Mean)	Crack Opening (µm)		Crack Number (Mean)
	Average	Standard Deviation		Average	Standard Deviation		Average	Standard Deviation	
7	16.5	3.4	8.5	31.6	9.3	0.9	21.7	5.4	2.8
9	17.9	4.8	7.5	44.5	27.5	1.8	44.2	7.1	1
14	17.6	5.2	3.4	56.3	16.2	1.1	21.6	8.2	2.6
15	15.5	4.4	5.8	34.3	8.6	1.1	18.9	7.8	2.8
16	17.1	4.5	4.6	49.8	13.7	1.0	22.9	8.4	2.3
19	16.0	3.9	3	45.8	13.7	1.2	27.1	7.1	1.4
20	13.5	2.4	6.4	16.0	2.6	0.3	0	0	0

**Table 3 materials-16-02576-t003:** CO_2_ emissions per slab’s constituent.

Concrete	Fibers	Steel Reinforcement
250 kg/m^3^	2.425 kg/t	1.932 kg/t

**Table 4 materials-16-02576-t004:** CO_2_ emissions per technical solution.

Technical Solution Number	7	9	14	15	16	19	20
CO_2_ (t)	57.8	63	57.7	57.8	57.7	63.0	39.4
Gain (%)	−8	0	−8	−8	−8	0	−37

## Data Availability

Not applicable.

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
