# Peer review of "Numerical Designing of Fiber Reinforced Concrete Eco-Constructions"

_materials, 2023, doi:10.3390/ma16072576_

Round 1

Reviewer 1 Report

"Numerical designing of fiber reinforced concrete eco-constructions" is manuscript on design methods and their standardization.

The author shows numerical models for cracking of fiber reinforced concrete (diffused cracking models, explicit cracking models - the cohesive crack models and the probabilistic explicit cracking model). He shows his concept on the example of geometry and dimensions of the Railway Track Mockup.

The article does not contain great scientific content, but it should be the basis for a constructive discussion for civil engineers. Personally, I believe that the presented concept is correct, although not the only one.

For this reason, I believe it should remain in this Journal.

Positive remarks:

I agree with the conclusion that: "the great conservatism of the construction industry for whom the Eurocodes and all other standards constitute a sort of inviolable pattern".

Critical remarks:

The author mainly develops his own concepts presented since 1987 (14/33 references are by Prof. Pierro Rossi).

Move title of chapter "5. Conclusion" to page 11.

Text formatting needs to be improved.

Author Response

Critical remarks:The author mainly develops his own concepts presented since 1987 (14/33 references are by Prof. Pierro Rossi).

I agree that the number of references related to my work is  a lot, but it is difficult for me to decrease it because, in the field of probabilisic models related to FRC cracking, I am a little beat alone. In the corrected version of the paper 10 references of my work are mentioned. 

Reviewer 2 Report

- Self-citations can be a legitimate way to reference earlier findings, but an abuse of them may sometimes be understood as an undue attempt to inflate an individual's citation count.  To avoid such a situation, please limit self-citations to a reasonable 10-15%.%

- The author should mention other ways to be circular and reach an eco-construction target, but from a material point of view. I suggest to cite for instance: Meza, Alejandro, et al. "Mechanical optimization of concrete with recycled pet fibres based on a statistical-experimental study." Materials 14.2 (2021): 240. Are those other strategies complementary with the numerical tool proposed? 

- authors should better enphasis the research significance of the research carried out, and the most relevant contributions in terms of novelty

- the author should better work and give more importance to section 4

Author Response

  • Self-citations can be a legitimate way to reference earlier findings, but an abuse of them may sometimes be understood as an undue attempt to inflate an individual's citation count.  To avoid such a situation, please limit self-citations to a reasonable 10-15%.%

I agree that the number of references to my work are a lot of but it is difficult to decrease it in the context of this paper. I am a little beat alone to develop probabilistic models for FRC cracking. In the revised paper I have limited the number of self-citations to 10. 

  • The author should mention other ways to be circular and reach an eco-construction target, but from a material point of view. I suggest to cite for instance: Meza, Alejandro, et al. "Mechanical optimization of concrete with recycled pet fibres based on a statistical-experimental study." Materials 14.2 (2021): 240. Are those other strategies complementary with the numerical tool proposed? 

The paper is not about all strategies to reach eco-construction target but only on the importance of numerical models to reach it.

  • authors should better enphasis the research significance of the research carried out, and the most relevant contributions in terms of novelty

I do it in the revised paper. I think that you are fully rigth.

  • the author should better work and give more importance to section 4

I do it in the revised paper.

Reviewer 3 Report

REVIEW

on article

Numerical designing of fibre reinforced concrete eco-constructions

Pierre Rossi

SUMMARY

The article submitted for review is devoted to a relevant topic. The issue of designing eco-structures made of fibre reinforced concrete (FRC) is considered. An analysis of the current state of the issue of designing eco-structures made of FRC was carried out. Design methods are considered, the current state of this problem is assessed, the author's reasoning is presented, supported by an analysis of existing experience. The article is of particular interest to readers who deal with the design of FRC structures, issues of eco-structures, as well as modern standards for the design of buildings and structures, including in European countries. Thus, the article can be useful for engineers and scientists involved in improving design standards, but the article has a number of serious shortcomings. The following are the comments that the reviewer considers important shortcomings.

COMMENTS

1.      The abstract of the article does not meet the requirements. The author only listed what he did in order to achieve the results of his research. The abstract should first of all contain a scientific problem, that is, the author assumes that the scientific problem is that the only way to design eco-structures from FRC is to use numerical experiments as a finite element method. This is probably the relevance of the study, and the problem of the study should be that there are unsolvable problems in the design or in approaches to the design of eco-structures made of FRC. Probably, there are no systematized norms, and that is why the author conducts his research. That is, the abstract should contain a scientific problem.

2.      In addition, the author lists the achievements that he received in the course of the study, but does not formulate the scientific result. The scientific result should be formulated more specifically in the abstract.

3.      In addition, the size of the abstract confuses. The fact is that the requirements for abstracts of this kind in the MDPI journal are about 200 words. The author used much less, thus the abstract does not reflect and does not reveal the essence of the article. Thus, the abstract needs serious processing.

4.      Keywords may need to be extended to 5 words so that such a study can be found by all interested engineers and scientists. The author is invited to complete the keywords.

5.      The introduction was made by the author in an overly journalistic style. Probably, the style of presentation should be changed somewhat from journalistic to scientific. For example, on line 53 there is the phrase “This is unfortunately not the case!”, therefore, the authors need to support their formulations with references to references. If this is the author's text, then I would like to give it a more structured look.

6.      The relevance of the research, its scientific novelty and practical engineering significance should be clearly formulated. The author should rework the Introduction section.

7.      In addition, I would like to see the program of the study. If this study is a reflection on the design of eco-structures from FRC, then I would like to see the main milestones: what scientific and engineering literature the author considers, what norms he compares, and perhaps this should be presented in the form of a flowchart.

8.      Section 2 is very short and does not reveal the purpose that it pursues. Perhaps the issue of the mechanical behavior of FRC eco-structures in tension should be highlighted in the Introduction subsection, because in fact in section 2 there is a literature review of 8 references.

9.      Section 3 "Numerical models for cracking of FRC structures" is built on the division into subsections and, in general, is quite hard to read. Perhaps the author should somehow restructure this section, and, as far as the reviewer understands, this section is one of the main results. Perhaps the author should restructure sections 3 and 4, and title the resulting one section "Results and Discussion", in order for the article to take on a more typical structure characteristic of research in the MDPI journal.

10.  There are comments on the Figure 1. The quality of the figure is poor. It should be presented in a higher quality. In addition, the figure is poorly explained.

11.  The same remark can be attributed to Figure 2. There are technical remarks in the figure: Figure 2 occurs 2 times, that is, there is a typo on line 308 and “Figure 3” should be put.

12.  In addition, the reviewer draws attention to the fact that in some figures the author writes Figure 1 in full, and in others he abbreviates Fig. 2. The author should bring the captions to the figures to uniformity.

13.  In general, the study is filled with a large number of typos and shortcomings, serious editorial work should be done.

14.  Figures 3 on lines 333 and 336 are presented in very poor quality.

15.  Also in tables 3 and 4 there are poorly readable characters. The authors should seriously work on the design of the article.

16.  The discussion of the results obtained is poorly done. You need to provide a clear, detailed justification of your point of view, based on design standards, existing cases in the design of FRC structures, and also compare your result with the results of other authors. If there are no studies on this topic and the discussion is relevant today, in any case, one should somehow refer to previous studies that dealt with similar topics. In particular, there are works, for example, in China, Europe, etc.:

https://doi.org/10.3390/ma15134697

https://doi.org/10.1016/j.istruc.2022.12.094

https://doi.org/10.1016/j.conbuildmat.2022.130075

https://doi.org/10.1016/j.jclepro.2022.132582

https://doi.org/10.3390/ma15134450

17.  As for the conclusions, they are also presented too superficially. I would like to see a clear formulation of the scientific result, the practical result, that is, the engineering value of the article, and a statement of what new knowledge was obtained or what existing ideas were developed.

18.  The list of references needs to be supplemented with 10-15 ref. on the design of eco-structures made of FRC over the past 5 years, because design standards and design issues are developing quite intensively, so scientific novelty should be justified.

19.   In general, the reviewer's remark on the article is as follows: the topic is interesting. It is obvious that the author is a professional in the design of eco-structures made of FRC. However, in order for the study to be published in the journal CivilEng, it should be very seriously finalized. The author is invited to make all the changes made by the reviewer and send the article for re-review.

Author Response

  1. The abstract of the article does not meet the requirements. The author only listed what he did in order to achieve the results of his research. The abstract should first of all contain a scientific problem, that is, the author assumes that the scientific problem is that the only way to design eco-structures from FRC is to use numerical experiments as a finite element method. This is probably the relevance of the study, and the problem of the study should be that there are unsolvable problems in the design or in approaches to the design of eco-structures made of FRC. Probably, there are no systematized norms, and that is why the author conducts his research. That is, the abstract should contain a scientific problem.

In my revised paper, I rewrite the abstract to consider this remark.

  1. In addition, the author lists the achievements that he received in the course of the study, but does not formulate the scientific result. The scientific result should be formulated more specifically in the abstract.

In my revised paper, I rewrite the abstract to consider this remark.

  1. In addition, the size of the abstract confuses. The fact is that the requirements for abstracts of this kind in the MDPI journal are about 200 words. The author used much less, thus the abstract does not reflect and does not reveal the essence of the article. Thus, the abstract needs serious processing.

I tried to improve this abstract to be more precise and attractive.

  1. Keywords may need to be extended to 5 words so that such a study can be found by all interested engineers and scientists. The author is invited to complete the keywords.

I have completed the keywords in the revised paper.

  1. The introduction was made by the author in an overly journalistic style. Probably, the style of presentation should be changed somewhat from journalistic to scientific. For example, on line 53 there is the phrase “This is unfortunately not the case!”, therefore, the authors need to support their formulations with references to references. If this is the author's text, then I would like to give it a more structured look.

I have changed the introduction to be more precise and scientific

  1. The relevance of the research, its scientific novelty and practical engineering significance should be clearly formulated. The author should rework the Introduction section.

I did it in the revised paper.

  1. In addition, I would like to see the program of the study. If this study is a reflection on the design of eco-structures from FRC, then I would like to see the main milestones: what scientific and engineering literature the author considers, what norms he compares, and perhaps this should be presented in the form of a flowchart.

I tried to do it in the revised paper.

  1. Section 2 is very short and does not reveal the purpose that it pursues. Perhaps the issue of the mechanical behavior of FRC eco-structures in tension should be highlighted in the Introduction subsection, because in fact in section 2 there is a literature review of 8 references.

In the revised paper Section 2 disappears and merges with Section 3

  1. Section 3 "Numerical models for cracking of FRC structures" is built on the division into subsections and, in general, is quite hard to read. Perhaps the author should somehow restructure this section, and, as far as the reviewer understands, this section is one of the main results. Perhaps the author should restructure sections 3 and 4, and title the resulting one section "Results and Discussion", in order for the article to take on a more typical structure characteristic of research in the MDPI journal.

I have restructured this section but I have merged Sections 2 and 3 and not 3 and 4. I think that it is better for the paper.

  1. There are comments on the Figure 1. The quality of the figure is poor. It should be presented in a higher quality. In addition, the figure is poorly explained.

I know that this figure is not completely satisfactory but I can do better presently. Moreover, it does not have a major relevancy for the understanding of the article. It's just informative.

  1. The same remark can be attributed to Figure 2. There are technical remarks in the figure: Figure 2 occurs 2 times, that is, there is a typo on line 308 and “Figure 3” should be put.

All is corrected in the revised paper. Concerning the quality of Figure 2, I can give the same answer than for Figure 1.

  1. In addition, the reviewer draws attention to the fact that in some figures the author writes Figure 1 in full, and in others he abbreviates Fig. 2. The author should bring the captions to the figures to uniformity.

That is corrected in the revised paper.

  1. In general, the study is filled with a large number of typos and shortcomings, serious editorial work should be done.

I hope that is corrected in the revised paper.

  1. Figures 3 on lines 333 and 336 are presented in very poor quality.

Same answer than for Figures 1 and 2.

  1. Also in tables 3 and 4 there are poorly readable characters. The authors should seriously work on the design of the article.

The Tables have been completely remade.

  1. The discussion of the results obtained is poorly done. You need to provide a clear, detailed justification of your point of view, based on design standards, existing cases in the design of FRC structures, and also compare your result with the results of other authors. If there are no studies on this topic and the discussion is relevant today, in any case, one should somehow refer to previous studies that dealt with similar topics. In particular, there are works, for example, in China, Europe, etc.:

https://doi.org/10.3390/ma15134697

https://doi.org/10.1016/j.istruc.2022.12.094

https://doi.org/10.1016/j.conbuildmat.2022.130075

https://doi.org/10.1016/j.jclepro.2022.132582

https://doi.org/10.3390/ma15134450

I am sorry to insist on the fact that it was the first time that FRC track slabs were designed and optimized by considering cracking at the service state. It was also the first time that a numerical model was used for this type of optimization.  

  1. As for the conclusions, they are also presented too superficially. I would like to see a clear formulation of the scientific result, the practical result, that is, the engineering value of the article, and a statement of what new knowledge was obtained or what existing ideas were developed.

I think (hope) I have improved this conclusion to propose you clear information.

  1. The list of references needs to be supplemented with 10-15 ref. on the design of eco-structures made of FRC over the past 5 years, because design standards and design issues are developing quite intensively, so scientific novelty should be justified.

I do not know others works concerning the design of FRC eco-construction using the same strategy as presented in this paper.

  1. In general, the reviewer's remark on the article is as follows: the topic is interesting. It is obvious that the author is a professional in the design of eco-structures made of FRC. However, in order for the study to be published in the journal CivilEng, it should be very seriously finalized. The author is invited to make all the changes made by the reviewer and send the article for re-review.

I tried to do everything possible to improve my article following your remarks.

Round 2

Reviewer 2 Report

.

Author Response

I tried to improve the paper and correct the english style.

Thank you for your help

Reviewer 3 Report

All my comments were considered and corrections were done. Just a few comments:

1. The author is invited to finalize the paper with the journal template.

2. Please, check all figures for the journal requirements: 1000 pix for the shortest side and 300 dpi.

3. The Abstract still needs the improvement. The journal requirements are:

The abstract should be a total of about 200 words maximum. The abstract should be a single paragraph and should follow the style of structured abstracts, but without headings: 1) Background: Place the question addressed in a broad context and highlight the purpose of the study; 2) Methods: Describe briefly the main methods or treatments applied. Include any relevant preregistration numbers, and species and strains of any animals used. 3) Results: Summarize the article's main findings; and 4) Conclusion: Indicate the main conclusions or interpretations. The abstract should be an objective representation of the article: it must not contain results which are not presented and substantiated in the main text and should not exaggerate the main conclusions.

Author Response

I tried to improve my abstract as you want.

Thank you for your help
